# Frugal and Translatable [^15^O]O_2_ Production for Human Inhalation with Direct Delivery from the Cyclotron to a Hybrid PET/MR

**DOI:** 10.3390/diagnostics14090902

**Published:** 2024-04-25

**Authors:** Jeffrey Corsaut, Elmer Soto, Heather Biernaski, Michael S. Kovacs, Keith St. Lawrence, Justin W. Hicks

**Affiliations:** 1Lawson Health Research Institute, London, ON N6A 5W9, Canadamkovacs@lawsonimaging.ca (M.S.K.);; 2Saint Joseph’s Health Canada London, London, ON N6A 4V2, Canada; 3Department of Medical Imaging, Western University, London, ON N6A 3K7, Canada; 4Department of Medical Biophysics, Western University, London, ON N6A 3K7, Canada

**Keywords:** positron emission tomography, GMP, oxygen-15, radiation safety, cerebral metabolic rate of oxygen, hybrid PET/MR

## Abstract

Oxygen-15 (β+, t_1/2_ = 122 s) radiolabeled diatomic oxygen, in conjunction with positron emission tomography, is the gold standard to quantitatively measure the metabolic rate of oxygen consumption in the living human brain. We present herein a protocol for safe and effective delivery of [^15^O]O_2_ over 200 m to a human subject for inhalation. A frugal quality control testing procedure was devised and validated. This protocol can act as a blueprint for other sites seeking to implement similar imaging programs.

## 1. Introduction

For insight into the role of altered brain energetics, positron emission tomography (PET) imaging of cerebral oxygen metabolism has been a vital tool [1,2]. For fifty years, PET has been the gold standard to quantitatively measure the cerebral metabolic rate of oxygen consumption (CMRO_2_). Originally, these measurements required three radioactive species containing oxygen-15 (^15^O; half-life 122s, 100% β^+^) [3,4]. The most obvious is [^15^O]O_2_ but to account for cerebral blood flow and volume, [^15^O]H_2_O and [^15^O]CO were required. This also required invasive arterial blood sampling for each radiotracer to account for recirculating [^15^O]H_2_O present plasma and [^15^O]O_2_ or [^15^O]CO bound to red blood cells. Recent work by our group and others has demonstrated that using a hybrid PET/magnetic resonance (MR) system permits measuring CMRO_2_ using only [^15^O]O_2_ [2,5,6,7]. This greatly reduced the experimental complexity and eliminated the need for arterial blood sampling. Additionally, MR-only measurements of CMRO_2_ could be simultaneously compared to quantitative PET measurements under identical experimental conditions using these hybrid imaging systems [8].

Nonetheless, the production of a single radiotracer was still required. At our facility, a stainless-steel delivery line carries the [^15^O]O_2_ approximately 202 m up six floors from the cyclotron facility to our hybrid PET/MR imaging equipment. Direct delivery of [^15^O]O_2_ for inhalation is not a new experimental technique [9], but this distance presented a unique challenge. We have devised a pre-release testing and delivery protocol adapted from previously successful strategies for the Current Good Manufacturing Practices (cGMP) manufacturing of [^13^N]ammonia [10] and [^15^O]water [11]. Although inline or parallel testing have been reported [9,12], the systems are technically complex, expensive, and could not be operated within close proximity of the PET/MR. By establishing a robust pre-release testing protocol, existing equipment could be adapted and safe inhalation of [^15^O]O_2_ can be achieved. This production method has been validated in porcine subjects [7] and now adheres to cGMP [13]. Human inhalation studies are currently underway, and results will be reported in due course.

## 2. Materials and Methods

### 2.1. Oxygen-15 Production and Delivery to PET/MR

Using the onsite PETtrace 800 cyclotron (16.5 MeV; GE Healthcare, Chicago, IL, USA), [^15^O]O_2_ was generated by the ^14^N(d,n)^15^O reaction from 1% O_2_ in N_2_ (both gases > 99.9999% purity). Bombardments were 1 min in length at 10 μA for quality control (QC) and for simulated patient doses during process validation. For target saturation yield, 6 min, and 40 µA bombardments were used to prepare ~40 GBq. [^15^O]O_2_ gas from the target flowed through a column of activated charcoal and ascarite to remove any ^15^O- or ^11^C-labeled CO or CO_2_, respectively. Radioactive gas was transferred to either the quality control lab or PET/MR by stainless steel lines with a target gas carrier.

If delivering to patients, a holding delay in the target was used to ensure roughly 2000 MBq would reach the scanner. This time delay was calculated for every two patient doses based on the QC dose calibrated amount. The delivery line to the PET/MR had an outer diameter of 3.175 mm, wall thickness of 0.7112 mm (radius = 0.08763 mm), and length of 218 m for a volume of 526 cm^3^. The target gas eluted at 800–1000 mL min^−1^ for 2 min. Bolus of [^15^O]O_2_ arrived at the PET/MR approximately 60 s after target opening. As a portion of the delivery line was within an outdoor conduit, integrity was tested by pressure hold once a quarter. During the cold months, the line was flushed with N_2_ prior to ^15^O production to ensure ice had not formed. Occupational radiation dose estimates were calculated at various points along the pathway using several calibrated survey meters to ensure building occupant safety.

### 2.2. Oxygen-15 Quality Testing

A two-position Valco Gas Chromatograph (GC) switching valve (Valco Instruments Company Incorporated (VICI), P/N EUD-64UWE, Houston, TX, USA) was used to isolate an aliquot of the target gas in a 0.5 cm^3^ Tefzel tube. In the first position (A), the target gas passed through the loop to the dose calibrator. In the second position, the helium carrier gas emptied the sample loop onto the GC (Figure 1). Dose calibration was completed using calibrated Capintec PET15 equipment (DC). The entire target effluent gas was trapped inside an emptied and dried 1 L sterile water bag (Figure 2). A 30 cm Tefzel tube (3.2 mm id) was attached to the bag to connect to a homemade, dual three-way solenoid device (designated as NAM) [10]. The outlet of the GC sampling loop on the VICI valve was connected to the common port of the first solenoid. The capture bag in the DC was attached to the normally open port on the solenoid. To trap the radioactive gas in the sampling loop, the normally closed port of the first solenoid was attached to the normally open port of the second solenoid. An exhaust line was attached to the common port of the second solenoid and led to a gas capturing system for decay. After opening the cyclotron target, gas was directed into the DC capture bag until the radioactivity plateaued. Triggering the solenoids via manipulator arms pressing a button trapped an aliquot of gas within the coiled tubing. Radioactivity was recorded along with the time to the second. The gas was held for approximately 2 min to determine the radionuclidic identity and purity of ^15^O by half-life determination.

Radiochemical identity and purity were determined using an SRI 310C GC (Mandel Scientific, Guelph, ON, Canada) equipped with thermal conductivity (TCD) and high sensitivity NaI gamma detectors, (Carroll and Ramsey 105S, Knoxville, TN, USA) in series. Gas components were separated on an SRI Molecular Sieve 31X column (180 cm of 3.2 mm id stainless-steel, Mandel Scientific, Guelph, ON, Canada) with a helium carrier gas flowing at 20 mL min^−1^. A sampling loop was placed in line between the cyclotron target and the dose calibrator gas capturing coil. While awaiting the half-life determination, the VICI valve was actuated to direct the target gas aliquot into the GC. Data were captured and analyzed using Chromperfect software (Justice Innovations, version 6.0.10). 

### 2.3. GMP Validation

Similar to previous short-lived isotope productions in our facility [10,11], a pre-release testing protocol was validated for up to two patient doses. A sequence of three batches was performed to represent a QC, dose 1, and dose 2 protocol. For the validation campaign, quality control was conducted on each batch in the sequence to demonstrate consistency. This process was repeated in triplicate. No sterility or stability testing was completed.

## 3. Results and Discussion

One of the most challenging aspects of this work was quantifying the target yield to deliver a safe level of radioactivity to the patient. After initially trying to use latex balloons placed in the DC, we found they were not strong enough to withstand the rapid inflation from the target emptying. Given the volume of gas at room temperature was circa 1.6 L, we elected to use an emptied 1 L sterile water for injection bag. From other GMP productions, we often have partially used bags available which would otherwise be disposed of at the end of the day. By severing the non-septum connection, the plastic tubing that remained was sufficiently long and wide enough to accept a shortened 1 mL syringe barrel (Figure 2). By fixing the syringe with electrical tape, the bag was gas tight. The luer lock permitted a secure connection to the solenoid device for trapping gas. We did not test whether a needle punctured through the septum would remain in place or be gas tight. After emptying non-bombarded target gas into our newly established gas capture bag, we found that the thicker plastic was able to withstand the pressures where latex could not. 

From target saturation yield tests during installation, our PETtrace cyclotron will produce approximately 40 GBq of [^15^O]O_2_ gas from a 6 min, 40 μA bombardment (4 µAh). Similarly, we typically obtain 10 GBq of [^15^O]water 4 min after the end of the bombardment (EOB) [11]. Assuming a near quantitative conversion and correcting for decay, these water productions also suggest a 40 GBq target yield. As such, when trapping a 6 min bombardment into the gas capture bag, we obtained 38.7 ± 2.1 GBq of radioactivity (decay corrected to EOB). This was again consistent with our commissioning saturation yields and the previous [^15^O]water experiments.

Having established that our gas capture bag can quantify a large bombardment, we conducted experiments with a smaller, 1 min, 10 μA bombardment (0.17 µAh). The activity measured was 2.75 ± 0.08 GBq with a range of 2.65–2.88 GBq (n = 9; Table 1). These values were nearly double the estimated yield if a GBq/µAh value was used to estimate the radioactivity produced. If a 4.0 μAh bombardment produced 40 GBq, a 10 GBq/μAh relationship can be established. Following a similar approach for a 0.17 μAh bombardment, 1.7 GBq should have been produced. We attributed this discrepancy to the amount of ^15^O generated during the cyclotron particle beam tuning steps prior to the bombardment. This process typically takes 1 min and uses a 12 μA beam which accounts for the near doubling of [^15^O]O_2_ produced (0.2 µAh). While it may be possible to add the beam tuning activity to the overall bombardment (0.37 µAh = 3.7 GBq at EOB), the time taken for these automated steps can be inconsistent from day to day. Thus, we will test the total target radioactivity with each production under the conditions used to produce the human inhaled doses.

The other quality metrics of radiochemical identity and purity were measured using a gas chromatograph. We placed a sampling loop between the cyclotron target and the dual solenoid device. This loop consisted of a two-position, six port valve connecting the cyclotron, solenoid device, GC, and exhaust to a gas-capturing system (Figure 1). In the primary position (A), the target gas enters through port 2 to port 1. Port 1 is connected to port 4 with a 0.5 cm^3^ tube. Gas then flows through port 3 into the dual solenoid device for triggered trapping. Port 5 and 6 are connected to He gas and the GC, respectively. Once the solenoids are triggered, the amount of radioactivity in the loop remains constant as there is no longer an exhaust pathway. After measuring for two minutes, a second radioactivity measurement is made with the dose calibrator to calculate the half-life of the captured gas. This also decreases the background radioactivity within the hot cell (i.e., a lead lined cabinet). At this point, the VICI valve is switched to the secondary position (B) and the sample loop is flushed with He onto the GC.

After passing through the column and reaching the TCD, we were able to separate the 1% oxygen from the nitrogen in the target gas. No other signals were recorded on the TCD during the method development or the nine batches in process validation. The radiation detector was placed next to the TCD exhaust and surrounded by lead. This resulted in a 1.8 s (0.3 min) delay between the TCD and radiation signal. The sole radiation signal detected was [^15^O]O_2_ with a relative retention time within 2.4 ± 2.4% of stable oxygen. Calibration of the GC with gas standards for CO and CO_2_ was not performed due to the large separation of these chemicals using the recommended GC separation method. Given that the 1% oxygen could be detected, if no other signal on the TCD was detected, it was assumed any species were <1%. No carbon source was present along the pathway to generate ^15^O-labeled CO or CO_2_; however, there was a low probability of forming ^11^C-labeled CO or CO_2_ if protons are present in the cyclotron ion source due to the ^14^N(p,α)^11^C nuclear reaction. If this were to occur, the half-life values would fail. As it was, we obtained an average half-life of 123 ± 4 s. That is within 6% of the half-life of 122 s. 

As we only want 1.8–2.2 GBq of [^15^O]O_2_ delivered, a delayed delivery time was instituted before releasing the gas from the cyclotron target. This value is calculated from the measured activity during quality control testing. The target yield is decay corrected to EOB and then used to determine the delay time, including the transit time between the cyclotron and PET/MR. We found a 1 min delay plus the 1 min transit time was adequate for delivery of approximately 2 GBq of [^15^O]O_2_ gas. To confirm the delivered amount at the PET/MR room, we repositioned a dose calibrator, dual solenoid device, and capture bag into the equipment room adjacent to the scanner. After a 1 min, 10 µA bombardment, the target gas was held for 1 min and delivered. A web camera was installed to monitor the delivery of the activity to the dose calibrator and to remotely trigger the solenoid device to capture the [^15^O]O_2_ gas once activity plateaued. The values obtained from this experiment were in line with the delivered radioactivity into the quality control lab. 

Given the long delivery line, we also ensured the safety of building occupants along the pathway. Using non-decayed delivery of a 1 min, 10 µA bombardment (approximately 4 GBq), we estimated the dose at various points along the pathway. For the calculations, we assumed a 4 GBq point source was placed along the delivery line near any occupied spaces as a worst-case scenario. The radiation dose estimates to building occupants were determined based on distance from and material between the stainless-steel line. These estimates were all within the safe limits for non-nuclear energy workers (<1 mSv/year). We then used several calibrated survey meters to measure the exposure during an actual delivery. Given the high velocity of the gas passing through the line and rapid decay, it was not surprising that the measured dose was significantly less than the estimate. The highest measured dose rate along the pathway was outside the PET/MR equipment room where the exhaust gas was trapped. Over the total delivery time of two minutes, a sum of 0.43 ± 0.06 µSv was measured, which was less than half the 0.92 µSv predicted. The areas within offices measured between 0.023 and 0.040 µSv (maximums), which remained under the 0.042 µSv predicted.

For the delivery of [^15^O]O_2_ gas, close coordination with the imaging team is required. Cyclotron bombardment started 5 min prior to the desired delivery time to the PET/MR. This delivery time was set to correspond with a particular MR sequence to record complementary data simultaneously. After the beam tuning had completed and the “on-target” beam indicator was visible on the cyclotron, a phone call was placed to the PET/MR control room to announce the exact delivery time (1 min + the daily delay calculated from QC). Upon release, the gas travelled through the delivery line to a sealed facemask within the PET/MR (Figure 3). This mask was connected to a ventilator which drew in ambient air and collected exhalations into a holding tank for decay. PET data were recorded for five minutes. If required, a second delivery was coordinated in a similar manner after a suitable delay for the first inhalation to decay. Once the final delivery of [^15^O]O_2_ is complete, the holding tank can be emptied after ten half-lives (i.e., 20 min). More image acquisition details will be provided in forthcoming publications. 

A surprise issue arose during one delivery attempt during the cold Canadian winter. We experienced a blockage which did not allow any gas to leave the cyclotron target. Given that the blockage disappeared when the exterior temperature increased, we attributed the blockage to ice formation within the line. In addition to our regular pressure test for line integrity, we also flushed the line with nitrogen in the fall prior to the dropping temperatures. The line is then capped inside the PET/MR equipment room unless in use. This has prevented water vapour from entering and freezing inside. 

There are several limitations or shortcomings that deserve to be mentioned. First, an already constructed device using three-way valves was used [10], but an even simpler two-way valve would suffice. Secondly, as mentioned above we did not test for the presence of either CO or CO_2_ gas. Given the short inhalation time, low volume, and standard GC testing method, this was deemed an acceptable risk at our site. However, others may require incorporation of standards to validate the GC method. It should also be noted that a manual three-way valve is installed in the chemical processing cabinet to direct the target gas to the QC lab or PET/MR room. Thus, the operator must remember to physically turn the valve prior to delivering to the PET/MR. It is our standard operating procedure to leave this valve directed to the QC lab (outlet is inside a sealed hot cell) unless delivering to the PET/MR and to return it to this position prior to shutting down the cyclotron after production. Electronic control with interlocks could be added to improve safety and avoid misdirected gas flow.

## 4. Conclusions

In summary, a method suitable for safe delivery of [^15^O]O_2_ gas for human inhalation was devised to suit our unique facility that used minimal expensive or highly technical resources. The protocol is highly translatable and permits installing a delivery line from the cyclotron to a PET scanner over a significant distance. The rapid transit time resulted in limited occupational exposure to workspaces along the delivery path. Inline testing by trapping the entire target contents within the dose calibrator was devised. The inline approach also allowed for sampling an aliquot by GC in a simple-to-construct manner. Other PET imaging facilities should thus be enabled to install systems for testing and delivering [^15^O]O_2_ without needing to relocate or significantly invest in any equipment to establish a [^15^O]O_2_ imaging program. 

## Figures and Tables

**Figure 1 diagnostics-14-00902-f001:**
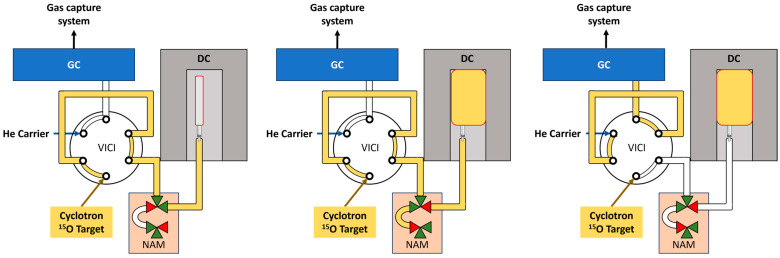
Connections and valve positions for capturing and testing [^15^O]O_2_ gas. LEFT: with the six-port valve (VICI) in position A, gas flows along the yellow pathway through the short sampling loop, through the three-way solenoid valve, and into a bag to capture the gas for dose calibration (DC). MIDDLE: the solenoids were activated and the [^15^O]O_2_ gas is trapped in the DC and sampling loop. RIGHT: while the gas is trapped in the DC, the VICI valve is actuated to position B and helium (He) carrier empties the small sample of gas onto the gas chromatograph (GC) for analysis. NAM = in-house designation for solenoid device [10].

**Figure 2 diagnostics-14-00902-f002:**
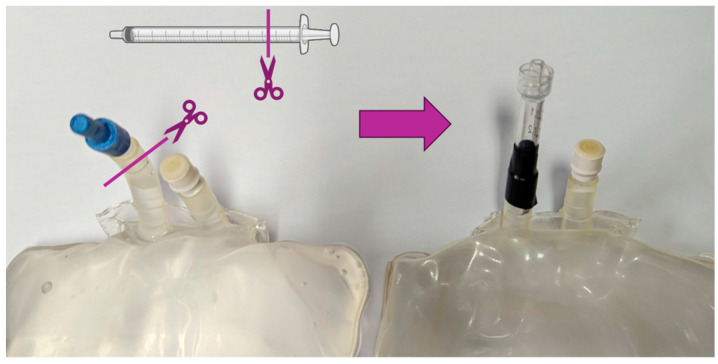
Adapting a 1 L sterile water bag to act as a gas capture bag for dose calibration of [^15^O]O_2_. Cuts are indicated for the syringe and bag with the produced gas capture bag to the right of the arrow. Using electrical tape, the bag could be pressurized and was gas tight.

**Figure 3 diagnostics-14-00902-f003:**
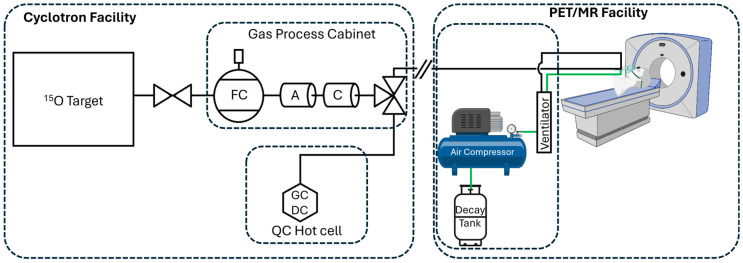
Schematic of gas delivery pathways from the cyclotron to the PET/MR equipment. Following bombardment, the ^15^O target is emptied through a flow controller (FC) at 800 mL min^−1^ to ascarite (A) and charcoal (C) columns to any potential [^15^O]CO_2_ and [^15^O]CO, respectively. A three-way valve directs the effluent gas to either a hot cell for quality control (QC) via gas chromatography (GC) and dose calibration (DC), or to the PET/MR facility. The [^15^O]O_2_ travels 200 m to a face mask connected to a ventilator which feeds in ambient air and collects the exhaled gases (green line). Exhaled air is collected via a compressor into a decay tank located in the adjacent equipment room. The hash mark between two facilities represents the stainless-steel delivery line which travels up the outside of the hospital (six floors) and about 100 m south of the cyclotron location.

**Table 1 diagnostics-14-00902-t001:** Summary of quality control results for all nine validation batches. EOB = end of bombardment; TCD = thermal conductivity detector.

Quality Metric	Specification	Result	
	Radiochemical Identity	t_1/2_ between 112 and 132 s	121 ± 4 s
	Radiochemical Identity	<10% difference in relative retention time	1.9 ± 1.6%
	Radiochemical purity	>95%	100%
Chemical Purity	No other detected peaks on TCD	Conforms	
	Measured Radioactivity	Between 2 and 3 GBq	2.75 ± 0.08 GBq
	(Corrected to EOB)	(4–6 GBq)	(5.40 ± 0.49 GBq)
	[^15^O]O_2_ delivered to PET/MR	Between 1.8 and 2.2 GBq	1.95 ± 0.31 GBq

## Data Availability

The original contributions presented in the study are included in the article, further inquiries can be directed to the corresponding author.

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
