# Peer review of "Frugal and Translatable [15O]O2 Production for Human Inhalation with Direct Delivery from the Cyclotron to a Hybrid PET/MR"

_diagnostics, 2024, doi:10.3390/diagnostics14090902_

Round 1
Reviewer 1 Report
Comments and Suggestions for Authors
Dear Editors,
The manuscript entitled: "Frugal and translatable [15O]O2 production for human inhalation with direct delivery from the cyclotron to a hybrid PET/MR" is a well-written communication paper describing a protocol for delivery of oxygen-15 from a cyclotron to a patient for inhalation and potentially performing PET acquisition. I do not have any major concerns to the manuscript. A few very minor remarks are listed below:
General remark: place citations, like [XX], before full stops/commas, in order to have it like this: (..... [XX]. or .... [XX],)
Line 25:
Is: 'rate of oxygen' should be: 'rate of oxygen consumption'
Line 49:
Please, explain the term 'QC'
Line 67:
Please, explain the term 'GC'
Line 75:
Please, explain the term 'VICI'
Line 136:
Please, explain the term 'EOB'
Generally, please explain the acronyms used on their first appearance.
Please, explain the acronyms used in figures and tables, in their subtitles.
Author Response
Thank you for taking the time to review our communication.
Citation notation has been updated throughout. Acronyms have been defined at first appearance in the text (sorry for that oversight). Acronyms are redefined if used in figure captions in case readers are skimming the paper.
VICI is a company name and the term has been moved to parentheses.
Reviewer 2 Report
Comments and Suggestions for Authors
Title: Frugal and translatable [15O]O2 production for human inhalation with direct delivery from the cyclotron to a hybrid PET/MR
Authors: Jeffrey Corsaut, et al.
Manuscript No. diagnostics-2930229
Production and delivery of short-lived PET isotopes for inhalation imaging studies enables specialized examinations with many applications. This study introduces a GMP-compliant practical solution to performing [15O]O2 inhalation studies with PET/MR.
This manuscript consists of a non-structured abstract with keywords, 4 sections (introduction, materials & methods with 3 subsections, results and discussion, and conclusions on 6 pages of single-spaced text with 2 embedded figures and 1 embedded table. There are 9 references, no appendices or supplements. No URLs are cited.
Delivery of cyclotron-produced [15O]O2 for PET/MR is a timely and novel innovation with important practical advantages and many potential applications. There are numerous technical and safety issues associated with this work that are clearly documented and explained in this report.
These authors have previously reported GMP of [15O]water for hybrid PET/MR studies in reference #9 (2020). Gas delivery has additional challenges that were met by this group. There is no illustration of the gas delivery path to the PET/MR scanner(s). Addition of this information would be helpful.
The regulatory requirements for this group in Canada have similarities to other nations. What, if any, additional requirements are imposed by FDA or other regulatory agencies? No FDA documents are cited. No example images are provided.
What instrumentation and procedures were used at the scanner for startup and shutdown of inhalation studies? How much scanner time was required to accomplish this? Were any safety issues encountered? The note of a "surprise issue" on page 5 is particularly valuable and interesting.
Overall, an experienced cyclotron physics and operations group introduced [15O]O2 for PET/MR with GMP compliance. This is an important practical solution to the technical and regulatory issues. This group is encouraged to introduce and highlight their prior GMP experience explained in reference #9, by citing this important reference in the Introduction. Several minor improvements to this report are suggested.
Author Response
Thank you for taking the time to review our manuscript. Your suggestions were helpful with greatly improving the overall product.
"This group is encouraged to introduce and highlight their prior GMP experience explained in reference #9, by citing this important reference in the Introduction."
Two sentences were added to the introduction to introduce the concept of pre-release testing and references our two prior publications conducting quality assessment with this strategy.
"There is no illustration of the gas delivery path to the PET/MR scanner(s). Addition of this information would be helpful."
A schematic has been added as Figure 3. The exterior pathway is difficult to map in 2D, as it travels upwards and horizontally across different sections of the hospital. Within the article, we mentioned the PET/MR is 6 floor above the cyclotron, and about 100 m south.
"The regulatory requirements for this group in Canada have similarities to other nations. What, if any, additional requirements are imposed by FDA or other regulatory agencies? No FDA documents are cited. No example images are provided."
Given the focus of most guidelines is upon parenteral injections, we adapted requirements for a PET gas to be as stringent as possible while working within the limitation imposed by gas handling and short half-life. To the best of our knowledge, radioactive gases, like O-15 or Xe-133, require (radio)chemical purity, identity, and radionuclidic purity assessments. Reference to the Canadian GUI-0071 for PET radiopharmaceuticals was added for comparison to individual jurisdictions.
"What instrumentation and procedures were used at the scanner for startup and shutdown of inhalation studies? How much scanner time was required to accomplish this? Were any safety issues encountered?"
We have added generic details regarding the delivery and shutdown. Further information will be included in a forthcoming paper on the imaging results.
To top of page 6 was added the following paragraph:
"For the delivery of [15O]O2 gas, close coordination with the imaging team is required. Cyclotron bombardment started 5 minutes prior to the desired delivery time to the PET/MR. This delivery time was set to correspond with a particular MR sequence to record complementary data simultaneously. After the beam tuning had completed and the “on-target” beam indicator was visible on the cyclotron, a phone call was placed to the PET/MR control room to announce the exact delivery time (1 min + the daily delay calculated from QC). Upon release, the gas traveled through the delivery line to a sealed facemask within the PET/MR (Figure 3). This mask was connected to a ventilator which drew in ambient air and collected exhalations into a holding tank for decay. PET data was recorded for five minutes. If required, a second delivery was coordinated in a similar manner after a suitable delay for the first inhalation to decay. Once the final delivery of [15O]O2 is complete, the holding tank can be emptied after ten half-lives (i.e. 20 min). More image acquisition details will be provided in forthcoming publications. "